# Serological Screening for Middle East Respiratory Syndrome Coronavirus and Hepatitis E Virus in Camels in Kazakhstan

**DOI:** 10.3390/pathogens11111224

**Published:** 2022-10-24

**Authors:** Kobey Karamendin, Aigerim Seidalina, Temirlan Sabyrzhan, Sardor Nuralibekov, Yermukhammet Kasymbekov, Symbat Suleimenova, Elizaveta Khan, Oralbek Alikhanov, Uldana Narsha, Kalya Erkekulova, Aidyn Kydyrmanov

**Affiliations:** 1Laboratory of Viral Ecology, Department of Virology, Scientific Production Center of Microbiology and Virology, 105 Bogenbay batyr Str., Almaty A25K1G0, Kazakhstan; 2Department of Veterinary Medicine, Agrarian Faculty, M. Auezov South Kazakhstan State University, 9th Corpus, 198 M.Kh. Dulati Str., Shymkent 160013, Kazakhstan

**Keywords:** camel, zoonosis, hepatitis E, MERS, serum, antibody, ELISA, Kazakhstan

## Abstract

After the recent Middle East Respiratory Syndrome coronavirus (MERS–CoV) pandemic in 2013, more attention has been paid to the camel as an important source of zoonotic viral infections. Almost simultaneously, in 2013, new genotypes 7 and 8 of the hepatitis E virus (HEV) were discovered in dromedary and Bactrian camels, respectively. HEV 7 was further shown to be associated with chronic viral hepatitis in a transplant recipient. In this study, serological screening for antibodies to MERS-CoV and hepatitis E virus was carried out on large camel farms in the south and west of Kazakhstan. 6.42% of the tested camels were found to be positive for antibodies to the hepatitis E virus, which indicates its circulation in local camel population. For the first time, antibodies to the hepatitis E virus were found in Bactrians, which have been little studied to date. Antibodies to MERS-CoV were not found in the camel sera.

## 1. Introduction

The *Camelidae* family, to which camels belong, is among the least studied domestic animals as a source of zoonotic infections. This family includes the genus *Camelus* (Linnaeus, 1758), consisting of two species: *C. dromedarius*, the dromedary or one-humped camel, and *C. bactrianus*, the Bactrian or two-humped camel. In Kazakhstan, the camel population is represented by both species, with a predominance of Bactrians. Camel breeding is a traditional industry in Kazakhstan and, according to the local statistics agency, 265,298 heads are being bred in Kazakhstan in 2022 [1]. From an epizootological point of view, in western Kazakhstan, there is an intensive import of dromedaries from neighboring Turkmenistan and Iran. Then, camels from these regions are often driven to the south of Kazakhstan. Thus, the possibility of introducing viruses from the Middle East is not excluded. Bactrian camels are mainly a local population in the south of Kazakhstan.

The discovery of the pandemic Middle East Respiratory Syndrome coronavirus (MERS-CoV) in 2012 and another novel camel coronavirus UAE-HKU23 caused an increased interest in the search for new pathogens in this animal [2,3]. Isolation of a novel MERS-related bat coronavirus NeoCoV, that uses human receptor ACE2, indicates the potential risks of the introduction into the human population [4].

Novel hepatitis E viruses (HEV) have been discovered in camels, which can serve as their potential reservoir and source of further transmission to humans. [5]. HEV is usually spread via the fecal–oral route. Hepatitis E viruses are transmitted from animals to humans through contaminated drinking water, by consumption of uncooked/undercooked meat or of organs from sick animals. Hepatitis E viruses belong to the *Hepeviridae* family, which includes two genera: *Orthohepevirus*, which infects terrestrial vertebrates, and *Piscihepevirus*, which infects fish. The *Orthohepevirus* genus consists of four species (A, B, C, D) with different host ranges. The Orthohepevirus A, which includes hepatitis E viruses, contains eight genotypes: HEV1 and HEV2, which have been found to be specific to humans, as opposed to HEV3 and HEV4, which have been recovered from humans and a number of other animal species such as swine, wild boar, and deer; HEV5 and HEV6, which circulate in wild boars; and HEV7 and HEV8, which were identified in dromedaries and Bactrians [6].

WHO research and statistics have shown that HEV caused 3.3 million cases of symptomatic acute hepatitis worldwide [7], and in some regions of the world, exceeded the more common hepatitis A prevalence [8]. Although the disease belongs to the self-limiting infections, high mortality rates were observed in infected pregnant women [7]. The only case of HEV7 detection in a human was reported in an immunocompromised patient with chronic hepatitis. [9].

Based on the data presented, we can assume that Bactrians, which are more common in Kazakhstan, can also serve as a potential reservoir of HEV in Kazakhstan. Serological testing of camels for antibodies to HEV has not been carried out previously in Kazakhstan. Cases of hepatitis E virus in humans are not officially registered in the Ministry of Health of Kazakhstan. Unfortunately, there are also no data on screening for HEV patients with undiagnosed hepatitis. Another important task was to check serologically whether the post-pandemic circulation of MERS-CoV exists in Kazakhstan camels to assess potential public health risks.

## 2. Materials and Methods

Serum samples were collected from the jugular vein of camels in five regions of Kazakhstan in 2020–2021 (Table 1). When choosing sample collection sites, we identified the regions with the most developed camel breeding facilities in Kazakhstan. In addition, studies on MERS coronavirus were previously conducted in the selected regions, allowing for longitudinal studies on MERS-CoV exposure. In each region, one farm with the highest concentration of camels of more than 50 heads was selected. In the western Mangystau and Atyrau regions, farms with dromedaries were selected due to their possible connection to the Middle East. The rearing methods differed only in the Mangistau region, where villages with a number of households with 2–3 camels prevailed. Sampling was carried out randomly in available households. In other regions, camels were reared in large farms up to 500 heads. From each large herd, 23–75 samples were selected randomly regardless of sex and age. When sampling, we did not consider the proportionality to the total number of camels in the region. From an epidemiological point of view, the Turkestan and Jambyl regions where dromedaries and Bactrians are reared in mixed herds are of particular interest. Rectal samples were collected using sterile swabs, which were subsequently placed in a viral transport medium containing antibiotics, antimycotics and 0.5% bovine serum albumin. In the field, samples were kept in liquid nitrogen and were stored at −70 °C after delivery to the laboratory.

Total anti-HEV antibodies were assessed in sera using HEV-Ab ELISA kit (ID Screen Hepatitis E Indirect Multi-species, ID Screen^®^, Grables, France), which is suitable for detecting anti-HEV antibodies in non-human sera.

Serum samples from camels were also tested with Recombivirus Camel Anti-Middle East Respiratory Syndrome Coronavirus (MERS-CoV) Nucleoprotein (MERS-NP) IgG ELISA kit (Alpha Diagnostic, San Antonio, TX, USA), which recognizes antibodies against MERS-CoV in camels. Both assays were performed according to the manufacturer’s instructions. A Multiskan FC analyzer (Thermo Fisher Scientific, Waltham, MA, USA) spectrophotometer was used to analyze the results. All dubious results were considered negative. As prescribed by the manufacturers, the final results were evaluated semi-quantitatively by calculating the ratio of the Optical Density (OD) value of the sample over the OD value of the Positive Control (HEV-Ab ELISA kit) or Calibrators (MERS-NP IgG ELISA kit,). A ratio greater than 0.7 (HEV-Ab ELISA kit) or 1.0 (MERS-NP IgG ELISA kit) was considered positive. Weighted prevalence was estimated using the SPSS 29.0 software (IBM, Armonk, NY, USA).

RNA extraction and RT-PCR. Viral RNA was extracted from 140 μL of viral transport media using the QIAamp RNA Mini kit (Qiagen, Hilden, Germany) following the manufacturer’s recommendations. Reverse transcription PCR (RT-PCR) assays were performed based on a one-step protocol using an appropriate One Taq One-Step RT-PCR kit (New England Biolabs, Ipswich, MA, USA) employing primers, targeting the conservative regions of HEV7 and HEV8 [5,10]. The final PCR products were visualized on 1% agarose gel.

## 3. Results and Discussion

### 3.1. Samples Collection

In 2020–2021, 249 blood serum samples were collected in the Turkestan, Mangystau, Jambyl, Atyrau and Kyzylorda regions that represent 86.4% of total camel population of Kazakhstan. (Figure 1). The age of camels ranged from 1 to 15 years. Camels under three years old were considered juveniles (Table 1).

### 3.2. Data Analysis

Sixteen of the 249 samples obtained from camels, 6.42% (95% CI 3.38–9.47%), were anti-HEV IgG-positive. A semi-quantitative evaluation of the reactivity against HEV has shown that those 16 positive samples had a ratio ranging from 0.71 to 1.48 (Appendix A). The seroprevalence among adult and juvenile camels were 2.64% (4/151) (95% CI 0.09–5.21%) and 16.0% (12/75) (95% CI 7.7–24.3%) respectively. A significant predominance among juveniles compared to adults was observed.

Of the five regions studied, the smallest proportion of seropositive to hepatitis E virus, 4.1% (2/48) (95% CI 1.49–9.82%), was registered in the Mangystau region. No antibodies were found in camels of the Atyrau region. It should be noted that Mangistau and Atyrau regions belong to the west of Kazakhstan, where only two seropositive samples were found out of 71 studied (2.8% (95% CI 1.03–6.67%)).

The most significant number of camels seropositive to the hepatitis E virus was registered in the Turkestan region, 27.0% (10/37) (95% CI 12.72–41.34%), which is in the south of Kazakhstan. In other southern regions, the seropositive prevalence was 6.0% (4/66) (95% CI 0.3–11.82%) for the Jambyl region and 0% for the Kyzylorda region. Thus, the weighted prevalence estimation confirmed the highest prevalence of antibodies to the hepatitis E virus in the southern Turkestan region. The presence of antibodies to the hepatitis E virus in bactrian camels was also registered, which was not previously described. Negative results were obtained with all 249 sera tested for antibodies to MERS-CoV. All the samples were also negative for HEV RNA, which was determined by conducting RT-PCR.

Hepatitis E was recognized as a disease of reemerging importance worldwide [11]. HEV prevalence among dromedary camels has been studied in African and Asian countries with intensive camel breeding practices. High seroprevalence values were observed in the African countries studied: Ethiopia (22.4%), Kenya (31.4%), Somalia (40.0%), Sudan (42.9%) and Egypt (62.9%) [12,13]. Comparably high values were found in Asian countries: the United Arab Emirates (37.1%), Pakistan (60.0%) and Israel (68.6%) [12,14]. A study of Bactrians conducted in Mongolia showed the absence of antibodies for HEV [13]. In Kazakhstan, we can see a much lower prevalence (6.42%) than in Asia and Africa. However, in the Turkestan region, a prevalence of 27% is higher than that in Ethiopia (22.4%), which indicates the significance of the hepatitis E virus, especially for the epidemiology of southern Kazakhstan. In this study, for the first time, seropositive cases were detected in Bactrian camels 7.4% (4/54) (95% CI 0.42–14.39%). However, it should be noted that in the camel breeding farm in the Jambyl region, both Bactrian and dromedary camels were reared together. Dromedaries in that region were seronegative.

A previous study in Ethiopia compared the antibody distribution in each age group. It was shown that the highest positive rate (76.4%) was in juveniles under three years old [13]. Studies in Israel have shown a prevalence of 67.4% in juveniles, but seroprevalence was also high in adult camels (88.9%) [14]. In our study, we observed 16% seroprevalence in juveniles, which is higher than that found in adults (2.64%). Previously, it was shown that all juveniles became infected during the first six months of life and cleared the virus after an average of two months [15]. This is the possible reason why negative HEV RNA results were obtained in seropositive camels. In general, previous studies have detected antibodies in camels up to 10 years of age and older. [14].

During the worldwide circulation of MERS-coronavirus in 2012–2015, cases of a disease in camels and humans were not registered in Kazakhstan. Serum samples from local camels were tested in 2015, and no seropositive cases were detected [16]. Further, in 2017 and 2018, during a serological screening of camels in Kazakhstan, positive cases were detected in 0.54% of bactrians and 0.24% of dromedaries out of 8207 examined serum samples [17]. This study did not reveal antibodies to MERS-coronavirus. The difference that was observed in the results of serological studies for MERS-CoV in Kazakhstan was possibly due to: (a) a significant period of time (two and four years) elapsing between the analyzes; (b) sampling being conducted in different herds; (c) different tests being used (a spike pseudoparticle neutralization test was used in 2015 that is less sensitive than the ELISA used in 2017–2018 and 2022); (d) the size of the tested population being considered (8207 sera in 2017–2018 against 550 in 2015 and 249 in 2021–2022). Anyway, the above-mentioned seropositive cases in camels in Kazakhstan indicate the potential for this infection. 

## 4. Conclusions

The potential for the spread of zoonotic infections from camels to humans in Kazakhstan was shown. In particular, 27% of camels in densely populated southern Kazakhstan being HEV-seropositive indicates the need for regular serological testing. Currently, people are not tested for the hepatitis E virus in Kazakhstan, and this article aims to draw attention to the problem in the healthcare system in regions with intensive camel breeding.

Our study has some limitations. Firstly, although we used the multi-species HEV-Ab ELISA kit suitable for animal sera, it is not standardized for camel sera testing. Also, this kit is designed to detect HEV3, but HEV7 is known to belong to the same serotype as HEV1–4 and actively cross-react with them [18]. It was shown that 90.0% of HEV7-positive sera reacted with the G3 antigen [13] indicating their high antigenic similarity. Second, a limited number of sampling locations were included in the study and extending the coverage of sampling areas is necessary. Additional sampling will be needed to reveal HEV infections in camels, especially in juveniles, and to determine the circulating HEV genotypes. Thirdly, further studies of human sera are needed, especially in the south of Kazakhstan, in order to assess the real epidemiological risks.

## Figures and Tables

**Figure 1 pathogens-11-01224-f001:**
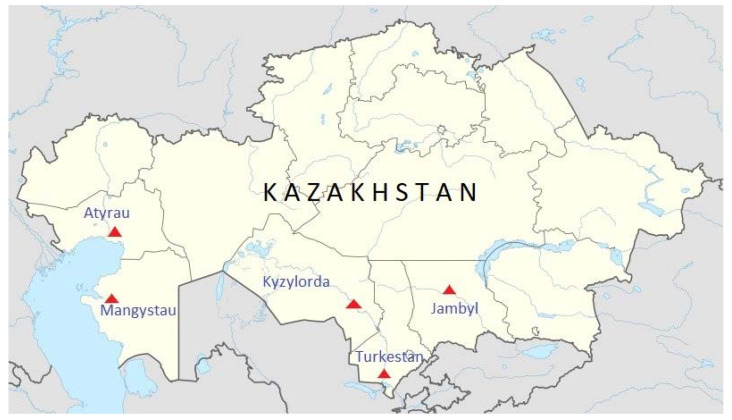
Camel sampling sites in Kazakhstan.

**Table 1 pathogens-11-01224-t001:** Characteristics of the collected serum samples tested for antibodies to hepatitis E virus.

Location	Camels in Region	Species	Age		Sex	Camels/Farms Tested	Positive		Weighted Prevalence, %
			Juv. ^2^	Adult			Juv.	Adult	
Mangystau	83,350	Dromedary	31	17	n.r. ^1^	48/1	1	1	1.5
Turkestan	38,543	Dromedary	33	4	n.r.	37/1	10	0	4.5
Atyrau	37,502	Dromedary	n.r. ^1^	n.r.	n.r.	23/1	0	0	0.0
Kyzylorda	61,592	Bactrian	3	72	73♀, 2♂	75/1	0	0	0.0
Jambyl	8309	Bactrian	5	49	51♀, 3♂	54/1	1	3	0.3
Jambyl		Dromedary	3	9	11♀, 1♂	12/1	0	0	0.0
Total	229,296		75	151		249	12	4	6.3

^1^ Not recorded, ^2^ juveniles.

## Data Availability

Not applicable.

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
