# Peer review of "Serological Screening for Middle East Respiratory Syndrome Coronavirus and Hepatitis E Virus in Camels in Kazakhstan"

_pathogens, 2022, doi:10.3390/pathogens11111224_

Round 1
Reviewer 1 Report
The authors of the study "Serological screening of camels for urgent viral zoonoses" did investigate the seroprevalence of two potential zoonotic pathogens recently identified in camelids, namely hepatitis E and MERS-CoV viruses in Kazakhstan in 2020-2022. A One-Health approach combining samplings in camelids and humans in the same regions of Kazakhstan should be promoted and would have bring more comprehensive information on HEV7/8 epidemiology.
The performances of the ELISA kits used in this study are poorly characterized in camels (the study investigating the kinetics of HEV7 infections markers in camels used a specific HEV7 ELISA), hampering the interpretation of the results obtained; alternatives should be found, the performances documented or the limits of the study should be discussed in the manuscript.
The manuscript needs to be modified according to the followings :
-specify the title to better reflect the screenings realized - only exposure to two viruses was investigated and the country should be added
-Abstract, line 15-16 : precise in which species the new HEV7 has been identified. Be careful not to overinterpret the findings on HEV7 and 8 : new genotypes of HEV have not been associated with increased risk of hepatitis in pregnant women, and only one human chronic HEV7 infection case has been described in immunocompromised patient. Please rephrase the sentence.
-Abstract, line 19-20 : please be careful here, there is a significant HEV seropositivity in camels but your study does not allow to infer any risk for the human population. Are transmission routes described for HEV7 and 8? Should you consider that HEV7/8 infection mostly occurs in young animals (as discussed later in the text), the consumption of milk is not a risk factor for human infections.
-Introduction, line 32 : would it be possible to screen the sera for this additional emerging coronavirus UAE-HKU23?
-Introduction, line 36-37 : please precise what is established about human risk of HEV7/8 infection
-Materials and methods, line 59 : the sampling period should be 2020-2021 according to line 80
-Lines 63 and 66 : the HEV indirect ELISA test uses HEV3 recombinant antigens and could the authors elaborate on the performances on such a kit to identify HEV7/8 infections? And could the positive sera found in the study correspond to false positives? The MERS-CoV NP ELISA kit is also designed for human sera and references on its performance to camel sera should be added to draw conclusions from the results obtained.
-Table 1 : the number of adults ad juveniles among tested specimens should be added. What do you consider as a juvenile?
How was the sampling performed (random sampling, other?): it is pivotal in the study and the interpretation of the results.
-Results, lines 91-92 : the authors are encouraged to provide a map of the regions tested with their seroprevalence
-Lines 103-104 : an additional sampling effort, in particular in juveniles, should be performed to establish HEV infections in camels through PCR screening of stools
-Lines 125-126 : please correct the mistake, no antibodies were detected in uninfected adult animals
-Lines 132-133 : how do you interpret such differences in MERS-CoV serosurveys along the years?
-Lines 134-135 : should be grouped with the paragraphs on HEV. Line 136, should be "10 months"
Reviewer 2 Report
The manuscript ""Serological screening of camels for urgent viral zoonoses" by Kobey Karamendin et al. provides data on the prevalence of two zonotic viruses in camels in different parts of Kazakhstan confirming the presence of Hepatitis E Virus in this region while Middle East Respiratory Syndrome Corornavirus is apparently absent in camels is Kazakhstan.
The study subject is of great interest and the data appear to be sound and the manuscript is nicely written. The reviewer encourages the authors to present some more details of the current data and, if possible, to conduct some additional analyses in order to strengthen the overall scientific merit of this study.
Major issues:
- Please modify the title by replacing “urgent” by “emerging” or to use both terms in combination.
- The authors should present the serological data together with the controls in order to allow a semi-quantitative evaluation of the reactivity detected against HEV.
- Do camelid antibodies allow distinction between early/initial and late phase of antigen exposition (comparable to early IgM and late(r) IgG in men or other mammals)? If so, a corresponding analysis would be helpful.
- The sentence in lines 129-131 is not really clear, in particular, the numbers of animals positive for MERS-CoV stated in the current manuscript and in the abstract of ref. 15 do not fit, please correct.
- Have pairs of juveniles / dams been tested and if so, how was the reactivity in the juveniles and dams?
- The authors should comment in some more detail on the abundance and spread of HEV in Kazakhstan and whether infections in the country have been already linked to contact to or transmission by camelids.
- The authors should comment on whether the camelid population in Kazakhstan is local/isolated or in exchange with areas where MERS-CoV or HEV are (more) abundant? In addition, features relevant to virus transmission among animals and to humans related to animal husbandry should be explained in more detail.
Round 2
Reviewer 1 Report
The authors considered most of the comments brought during the reviewing process and discussed more thouroughly their findings.
My comments are listed below :
Introduction, line 58-59 : only one case of chronic hepatitis associated with HEV7 infection has been reported in the literature. Please correct the sentence accordingly.
Line 65 : should be "hepatitis"
Material, lines 71-72 : "so the studies on the regular basis was also important." could be replaced by "allowing for longitudinal studies on MERS-CoV exposure".
Lines 70-74 : although precisions on collection sites were obtained, more details should be given on the selection of farms and of animals sampled in the selected farms. In particular, was the sampling randomized or not and how were the animals selected in each facility? Were the number of animals sampled in a specific region proportionate to the total dromedary population in the region?
Results, line 110 : replace "varied" by "ranging"
LInes 133-134 and more generally in the manuscript : please check references for consistencies. Reference 12 does not correspond to studies performed in Mongolia.
Line 158 : hypothesis a) refering to different sampling dates could be precised; one could infer that virus circulation and exposure occured between 2015 and 2017-2018. The size of the population is also one parameter that should be considered - the second survey conducted in 2017-2018 was also the largest one, allowing the identification of low MERS-CoV seroprevalences in the studied population.
Round 3
Reviewer 1 Report
The different modifications requested were brought by the authors. Sampling design was provided, and from my understanding on the sampling design, only a few farms were sampled per facility. The exact number of farms should be clearly indicated in table 1 and the limited number of sampling locations should also be discussed in the results and discussion part.
